# Salt-Controlled Vertical Segregation of Mixed Polymer Brushes

**DOI:** 10.3390/ijms252313175

**Published:** 2024-12-07

**Authors:** Ivan V. Mikhailov, Anatoly A. Darinskii

**Affiliations:** NRC «Kurchatov Institute»—PNPI—IMC, St. Petersburg 199004, Russia; i.v.mikhailov-imc.ras@yandex.ru

**Keywords:** mixed polymer brushes, height switch, phase segregation, strong polyelectrolyte

## Abstract

Using the self-consistent field approach, we studied the salt-controlled vertical segregation of mixed polymer brushes immersed into a selective solvent. We considered brushes containing two types of chains: polyelectrolyte (charged) chains and neutral chains. The hydrophobicity of both types of chains is characterized by the Flory–Huggins parameters $\chi_{C}$ and $\chi_{N}$, respectively. It was assumed that the hydrophobicity is varied only for the polyelectrolyte chains ($\chi_{C}$), while other polymer chains in the brush remain hydrophilic ($\chi_{N}=0$) and neutral. Thus, in our model, the solvent selectivity ($\chi =\chi_{C}-\chi_{N}$) was varied, which can be controlled in a real experiment, for example, by changing the temperature. At low salt concentrations, the polyelectrolyte chains swell and occupy the surface of the mixed brush. At high salt concentrations, the hydrophobic polyelectrolyte chains collapse and give place to neutral chains on the surface. By changing the selectivity of the solvent and the ionic strength of the solution, the surface properties of such mixed brushes can be controlled. Based on the numerical simulations results, it is shown how the critical selectivity corresponding to the segregation transition in polyelectrolyte/neutral brushes depends on the ionic strength of the solution. It is shown that at the same ionic strength, the critical selectivity increases with an increasing degree of dissociation of charged groups, as well as with an increasing fraction of polyelectrolyte chains in the mixed brush. It has also been shown that at low ionic strengths, the critical selectivity of the solvent decreases with increasing grafting density, while at high ionic strengths, on the contrary, it increases. Within the framework of the mean field theory, a two-parameter model has been constructed that quantitatively describes these dependencies.

## 1. Introduction

The development of smart surface coatings is one of the promising areas of modern nanotechnology. One of the most popular methods in this field is the use of polymer brushes as smart polymer coatings (SPCs), which can be used in many applications [1]. Examples are switch sensors [2,3,4,5], antifouling surfaces [6,7,8,9], lubrication [10], targeting drug delivery [11,12,13,14,15], and chromatographic protein separations [16,17,18].

To control the surface properties of brushes by changing external conditions, mixed brushes are used, consisting of polymer chains of different structures. An example is brushes consisting of chains with different affinities for the solvent. By changing the affinity for the solvent due to the solvent composition, the ionic strength, or the temperature of such brushes, a segregation of the grafted chains may occur. Extreme cases of segregation include lateral and vertical ones. If the brush components are highly incompatible, lateral segregation arises. Alternatively, if the compatibility between the brush components is high enough, vertical segregation occurs [19,20,21,22,23]. The greatest interest for smart coating creation is mixed brushes with vertical microsegregation, which implies that macromolecules of one type are located inside the brush near the grafting surface and macromolecules of another type form the periphery of the brush.

A characteristic feature of such brushes is a sharp change in the morphology of the brush in a narrow range of environmental parameters. In this case, a switching occurs: the chains from the surface go inside the brush, and the ends of the initially internal chains form a new surface of the brush with new properties. If one of the components is a polyelectrolyte, then another control parameter arises with which you can change the morphology of the brush: the ionic strength of the solution in which the mixed brush is immersed. If the brush consists of polyelectrolyte and neutral chains, then the change in ionic strength will affect mainly the polyelectrolyte chains. It is well-known that the size of the polyelectrolyte chains in a brush decreases with an increase in the ionic strength of the solution. If the structure of the mixed brush is such that at low ionic strengths the ends of the charged chains are higher than the ends of the neutral chains, then, as the ionic strength increases, the polyelectrolyte chains may appear lower than the ends of the neutral chains. Accordingly, the properties of the brush surface will change, and we observe a conformation transition similar to that for mixed brushes with different affinities.

Several experimental and theoretical works have considered brushes that contain only polyelectrolyte chains [24,25,26,27,28,29]. There are also several experimental studies of mixed brushes that contain both polyelectrolyte and neutral chains [30,31,32]. In terms of the theory of such brushes, there is only work [33], where polyelectrolyte and neutral chains can change positions on the grafting surface. In such brushes, a change in charge or ionic strength will lead to the lateral phase separation. This work is the first to consider a mixed brush in which vertical micro-segregation of charged and neutral chains is possible. In such a brush, the points of attachment of the chains to the surface are fixed, and when external conditions change, the positions of the free ends may change as well.

The main goal of our study is to determine the conditions under which the segregation transition occurs (i.e., when the charged chains on the surface of the brush are replaced by the ends of the neutral chains).

Polyelectrolyte brushes can be found in three regimes with respect to medium ionic strength: “osmotic”, “salted” and “quasi-neutral” regimes [25]. Under conditions of low ionic strength, the localization of counterions inside the brush leads to the generation of osmotic pressure (osmotic regime) that promotes the brush swelling. When the ionic strength increases to a value comparable to the value of the ion concentration inside the brush, a “salted brush” regime is observed. The addition of ions to the medium leads to a decrease in the osmotic pressure of counterions, and the brush begins to collapse due to entropic elasticity (in our case also due to hydrophobic interactions). At very high ionic strength, the polyelectrolyte brush goes into a quasi-neutral state, in which the influence of the ion-osmotic effect is negligible compared to the influence of the non-electrostatic interactions. If the solvent is poor under such conditions, the brush will collapse to the limit.

In a mixed brush (Figure 1), polyelectrolyte chains are “diluted” with neutral chains, but the same trends for the polyelectrolyte chains remain. If both neutral and charged chains have the same contour length and the charged ones have a worse affinity for the solvent, the position of the end units in the brush “switches”. At low ionic strength, the ends of the polyelectrolyte chains are located at the periphery of the brush, shielding the surface from neutral chains. At a high salt concentration, the opposite effect is observed: polyelectrolyte chains are collapsed, giving way to neutral chains on the periphery of the brush. At some intermediate salt concentration, there is a transition point. At this point, the end positions (chain heights $H_{C}$ and $H_{N}$) for the charged and neutral chains are at the same level. The segregation of such brushes can be controlled by varying two intensive parameters of the system: the ionic strength of the solution and the selectivity of the solvent (temperature). The purpose of this work is to predict the conditions under which the vertical segregation of mixed polyelectrolyte-neutral brushes occurs depending on the composition of the brush and the molecular parameters of the grafted chains.

## 2. Results and Discussion

### 2.1. Analytical Theory

For an analytical prediction of the critical value of the incompatibility of polyelectrolyte chains with a solvent χ, we have developed the theory based on the simple box-model [26]. In this model, the ends of the grafted chains are not distributed over the brush but are fixed at the same distance *H* from the grafting surface, and all chains are evenly stretched. For simplicity, the volume fraction of monomer units is considered as an average value independent of the distance from the grafting surface:(1)$$\varphi =\frac{N\sigma }{H},$$
where σ is the total grafting density of charged and neutral chains, and *N* is the degree of polymerization of these chains.

To find the value of $\chi^{*}$ corresponding to the transition point, we assume that the ends of both types of chains are located at the same distance $H=H_{0}$ from the grafting surface, where $H_{0}$ is the height of the neutral brush with the same total grafting density as in the mixed one.

In an equilibrium state, a balance of pressures (forces per unit grafting surface area) acting on the brush must be maintained [28]:(2)$$f_{cb}+f_{el}+f_{mix}+f_{FH}=0$$

Here, $f_{cb}$ is the osmotic pressure of mobile ions, $f_{el}$ is the negative elastic pressure due to chain stretching, $f_{mix}$ is the osmotic pressure caused by the entropy of polymer–solvent mixing, and $f_{FH}$ is the pressure causing hydrophobic compression of polyelectrolyte chains (contribution of the Flory–Huggins interaction energy).

The first electrostatic contribution can be expressed by charge balance (the so called Donnan equilibrium [26]):(3)$$f_{cb}=\frac{N\sigma X\alpha }{H}\left[\sqrt{1+y^{2}}-y\right],$$
where α and *X* are the degree of ionization of polyelectrolyte chains and their fraction in the composition of the mixed brush. Thus, the factor before the square brackets represents the concentration of charged monomer units in the brush. Parameter *y* denotes twice the ratio of salt concentration and charged monomer unit concentration:(4)$$y=\frac{2IH}{55.5XN\sigma \alpha }$$

The factor 55.5 is used to convert ionic strength (the concentration of salt ions) from mol/L units to a dimensionless volume fraction. Since one liter of water contains $c_{w}=55.5$ mol, the dimensionless volume fraction of salt ions is calculated as the ratio $\varphi_{s}=c_{s}/c_{w}=I/c_{w}$, where $c_{s}$ is the concentration of monovalent salt ($c_{s}=I$).

The second term in Equation (Equation 2) is the force that must be applied to stretch the freely jointed chain to the distance *H* between the grafted and free ends. In the regime of Gaussian stretching of chains, this contribution can be represented as
(5)$$f_{el}=-\frac{d}{dH}\left(\frac{3}{2}\sigma \frac{H^{2}}{N}\right)=-3\sigma \frac{H}{N}$$

The contribution of the entropy of polymer–solvent mixing, according to the Flory theory, is calculated by the formula:(6)$$f_{mix}=-\frac{d}{dH}\left(\frac{1}{2}\varphi^{2}H\right)=\frac{1}{2}{\left(\frac{N\sigma }{H}\right)}^{2}$$

Hydrophobic forces promoting the decomposition of polyelectrolyte chains with an increasing Flory–Huggins parameter are described by the following expression:(7)$$f_{FH}=-\frac{d}{dH}\left(-\chi X\varphi^{2}H\right)=-{\left(\frac{N\sigma }{H}\right)}^{2}\chi X$$

Taking into account all the equations described above, we obtain the desired equation that describes the dependence of the critical value $\chi^{*}$ on the structural parameters of the mixed brush and the ionic strength:(8)$$\chi^{*}=\frac{1}{X}{\left(\frac{H}{N\sigma }\right)}^{2}\left[f_{cb}+f_{el}+f_{mix}\right]$$

At the transition point, hydrophobic interactions neutralize electrostatic ones, resulting in the polyelectrolyte and neutral chains take on identical conformations, corresponding to the conformations in a neutral brush in an athermal solvent. In this case:(9)$$f_{mix}+f_{el}=0$$
and both types of chains are extended to the height $H=H_{0}$ of the brush with a total grafting density σ in an athermal solvent.
(10)$$H_{0}=N{\left(\frac{\sigma }{6}\right)}^{1/3}$$

Given these model assumptions, Equation (Equation 2) can be written as
(11)$$\chi^{*}=\frac{\alpha }{\varphi_{0}}\left(\sqrt{1+{\left(\frac{2I}{55.5X\alpha \varphi_{0}}\right)}^{2}}-\frac{2I}{55.5X\alpha \varphi_{0}}\right)$$
where $\varphi_{0}=N\sigma /H_{0}\approx 1.8\sigma^{2/3}$ is the mean density of monomer units in a neutral brush under athermal solvent conditions.

Two extreme cases can be considered. At high ionic strengths (and at σ→0 or X→0), the critical selectivity $\chi^{*}$ decreases inversely with increasing ionic strength *I*:(12)$$\chi^{*}\to \frac{55.5}{4}\cdot \frac{X\alpha^{2}}{I}$$

At low ionic strengths, the value of $chi^{*}$ ceases to depend on ionic strength:(13)$$\chi^{*}\to {\left(\frac{\sqrt{6}}{\sigma }\right)}^{2/3}\alpha$$

These predictions are scaling due to the primitiveness of the model; however, the presented analytical model suggests that the dependence (Equation 11) can be described as a two-parameter one using two empirical coefficients *A* and *B*:(14)$$\chi^{*}=A\left(\sqrt{1+{\left(BI\right)}^{2}}-BI\right)$$

### 2.2. SCF Simulation Data

The analytical theory uses a box model, which has limitations. In particular, this model assumes that the ends of the grafted chains are located at the same distance from the grafting surface.

The numerical SCF model does not have this limitation. As can be seen from Figure 2, the end groups are located throughout the volume of the mixed polymer brush.

Figure 2 shows how the “switching” of the position of the grafted chains occurs. In the middle column of the upper and lower graphs, the parameters are selected so that the free ends (for neutral and polyelectrolyte chains) are located equally relative to the grafting surface. This positions were determined in the work as the first moment of the distribution of the ends:(15)$$H=\frac{\sum_{z}g\left(z\right)z}{\sum_{z}g\left(z\right)}$$

As can be seen from the graphs, the end groups of the polyelectrolyte chain can occupy a place on the periphery of the brush or can go closer to the grafting surface, depending on the selectivity of the solvent χ or the ionic strength *I*. Thus, the segregation of chains in the brush can be controlled by changing the selectivity of the solvent (temperature) and the salt concentration (ionic strength of the solution). At each value of the ionic strength, it is possible to determine the critical selectivity of the solvent at which vertical segregation occurs.

Figure 3 shows the dependencies of the critical value of χ on *I* for different values of the fraction of polyelectrolyte chains in the mixed brush and different values of the degree of polymerization of the grafted chains in double logarithmic coordinates. This graph can be considered as a diagram. Each curve divides the parameter region $\left(I,\chi^{*}\right)$ into two parts: above the curve and below the curve. The parameter region below the curves corresponds to the state when the free ends of the neutral chains are located below the charged ones. And, vice versa, above the corresponding curves the ends of the polyelectrolyte chains are located below the neutral ones.

It can be noted that these dependencies reach a “plateau” in the region of low salt concentrations (in the “osmotic” regime of the brush). In other words, the value of the segregation point $\chi^{*}$ at the low ionic strength of the solvent tends to some constant value that does not depend on the fraction of polyelectrolyte chains in the brush. Thus, in the region of low salt concentrations, vertical segregation can occur only due to a change in the solvent selectivity over a wide range of ionic strength. As the salt concentration increases, the critical selectivity decreases inversely proportional to the value of the ionic strength of the solution. In this region of the salt concentration, a higher fraction of polyelectrolyte chains *X* corresponds to a higher critical selectivity $\chi^{*}$.

This conclusion is also consistent with the analytical theory (Equations (Equation 12) and (Equation 13)). The obtained dependencies also have the predicted form of the function (Equation (Equation 14)). Figure 3 shows the data fitting by the two-parameter model (Equation (Equation 14)) with dotted lines.

An increase in the degree of ionization of monomer units of the polyelectrolyte chains, according to our analytical theory, should lead to an increase in the critical selectivity of the solvent necessary for the segregation transition at any ionic strength of the solution. The numerical simulation data fully confirm this prediction (Figure 4).

In case of increasing grafting density σ, our analytical theory predicts a decrease in the value of the segregation transition point $\chi^{*}$ at low salt concentrations, which is also confirmed by the results of our numerical SCF simulation (see Figure 5).

However, in the region of high salt concentrations, a discrepancy between the analytical prediction and the modeling is observed. The value of $\chi^{*}$ increases with increasing grafting density σ, although, according to Formula (Equation 12), the value of $\chi^{*}$ should not depend on σ at high salt concentrations. This contradiction is explained by the non-uniform distribution of the density of monomer units and the charge distribution in the volume of the brush, which is not taken into account by the simplified box-model, but it is taken into account in the SF-SCF modeling. But even in this case, a two-parameter model (Equation 14) is applicable to describe the dependence $\chi^{*}\left(I\right)$.

## 3. Materials and Methods

### 3.1. Model

To study salt-controlled vertical segregation in mixed polymer brushes, we consider the following coarse-grained model (Figure 1). A flat brush is formed by two types of chains: neutral (uncharged) chains and charged (polyelectrolyte) chains. Their degrees of polymerization are $N_{N}$ and $N_{C}$, and their grafting densities are $\sigma_{N}$ and $\sigma_{C}$, respectively. The grafting density is a dimensionless quantity equal to the ratio $\sigma =a^{2}/s$, where *a* is the linear size of the monomer unit of the grafted chain, and *s* is the grafting surface area per grafted macromolecule. In other words, the grafting density σ is the number of chains per unit area of the grafting surface. The molar fraction of polyelectrolyte chains $X=\sigma_{C}/\left(\sigma_{C}+\sigma_{N}\right)$ is the ratio of the number of polyelectrolyte chains to the total number of chains in the brush. All chains are considered as flexible, and the size of the monomer unit *a* coincides with the statistical length of the segment. The chains are grafted at one end onto a flat impenetrable uncharged surface.

The brush is immersed into a sufficiently large reservoir of aqueous solution of salts with concentration $c_{s}=c_{+}=c_{-}$, which contains monovalent cations ($c_{+}$) and anions ($c_{-}$) carrying an elementary charge |e|. Thus, the molar ionic strength *I* of the salt solution is equal to the salt concentration:(16)$$I=\frac{1}{2e^{2}}\left(c_{-}z_{-}^{2}+c_{+}z_{+}^{2}\right)=c_{s}$$

The counterions are indistinguishable from free cations in solution. It is assumed that the monomer units, the ions, and the neutral water molecules have the same linear size *a*. We will also assume that this value is equal to the linear size of a water molecule: a=0.31 nm. Our simulations were performed at a fixed temperature T=298.15 K and at a fixed relative dielectric constant of the medium $\epsilon_{r}=80$ that corresponds to the Bjerrum length:(17)$$\lambda_{B}=\frac{\left|e\right|}{4\pi \epsilon_{r}\epsilon_{0}k_{B}T}\approx 0.7\;\mathrm{nm},$$
where $\epsilon_{0}$ is the vacuum permittivity and $k_{B}$ is the Boltzmann constant.

The Debye screening length in the bulk of the solution is determined by its ionic strength:(18)$$\kappa^{-1}={\left(8\pi N_{A}\lambda_{B}I\right)}^{-1/2},$$
where $N_{A}$ is the Avogadro constant. As the ionic strength decreases from I=1 M to $I=10^{-5}$ M, the Debye length increases from $\kappa^{-1}=a$ to $\kappa^{-1}=313a$. The bound charges on the polyelectrolyte chains cease to have an electrostatic effect on the free ions at a distance $z_{max}\gg \kappa^{-1}$ from the surface of the polymer brush, and the local ion concentration reaches the setup level concentration $c_{s}$. The boundary condition of the model is a fixed value of the concentration of the co-ions of charged chains ($c_{-}=c_{s}$) at a large distance ($∼z_{max}$) from the brush surface. The concentration of counterions is selected automatically during the modeling process based on the condition of electrical neutrality.

Charged chains in a mixed brush are considered to be strong polyelectrolytes. The fraction α of the ionized monomer units in a strong polyelectrolyte is independent of the pH and ionic strength of the solution. In other words, each charged chain in the brush has a fractional charge α per monomer unit. The sign of the charge of the polymer chain does not affect the simulation results, and it was chosen negative for definiteness.

Neutral and charged chains also have different affinities for the solvent, i.e., the solvent is selective with respect to different types of grafted chains. The hydrophobicity of the polyelectrolyte chains and the neutral chains in the brush is characterized by the Flory–Huggins parameters, $\chi_{C}$ and $\chi_{N}$, respectively. It is assumed that the solvent is athermal with respect to the neutral chains ($\chi_{N}=0$), but polyelectrolyte chains are capable of changing their hydrophobicity. The strength of selectivity is specified in the model as the difference $\chi =\chi_{C}-\chi_{N}$. The same definition of selectivity is used, for example, by the authors of the article [34]. Since $\chi_{N}=0$, the solvent selectivity is characterized by the incompatibility of water molecules and monomer units of charged chains.

The segregation transition point is taken to be a combination of parameters (I,χ), at which the first moments of the distribution of the end segments in the direction normal to the grafting surface for both types of chains are equal ($H_{C}=H_{N}$, see Figure 1).

### 3.2. Method

A one-gradient Scheutjens–Fleer self-consistent field (SCF) numerical method was used to study vertical segregation in mixed polyelectrolyte brushes. This method for studying polyelectrolyte systems was first proposed in the article [35] and then improved in the work [36].

The self-consistent field method uses the basic assumption of the mean-field approximation that the consideration of a system of many molecules can be replaced by the consideration of the conformations of a limited number of molecules of different types that are under the influence of a common effective field that describes their interaction with each other and with other molecules. The main idea of this assumption is to reduce the solution of the problem using particle coordinates to the problem of finding the distribution of the volume fraction of particles φ(z) in the field of a self-consistent potential u(z). Moreover, in the one-gradient method, these distributions are considered as functions of a single coordinate *z*, which is normal to the grafting surface of macromolecules.

To find the equilibrium distributions of $\varphi_{i}\left(z\right)$ for different types of molecules i∈A (where *A* is a set of different types of particles: charged and neutral monomer units, water molecules, and salt ions), the free energy of the molecular system is numerically minimized. This problem is solved by the method of Lagrange multipliers (Lagrange field λ(z)), taking into account the incompressibility condition:(19)$$\sum_{i\in A}\varphi_{i}\left(z\right)=1\;\;\;\forall z$$

The functional to be minimized is
(20)$$F\left[\varphi ,u,\lambda \right]=-\sum_{i\in A}\ln Q\left[u_{i}\left(z\right)\right]-\sum_{i\in A}\sum_{z}u_{i}\left(z\right)\varphi_{i}\left(z\right)+F_{int}\left[\varphi \right]+\sum_{z}\lambda \left(z\right)\left(\sum_{i\in A}\varphi \left(z\right)-1\right),$$
where $Q[u_{i}\left(z\right)]$ is the partition function of molecular component *i*, and $F_{int}$ is the free energy of the interaction between these components, consisting of the excluded bulk interaction and the Coulomb interaction:(21)$$F_{int}=\frac{1}{2}\sum_{z}\sum_{a}\sum_{b}\chi_{a,b}\varphi_{a}\left(z\right)\left(\langle \varphi_{b}\left(z\right)\rangle -\varphi_{b}^{bulk}\right)+\frac{1}{2}e\sum_{i\in A}z_{i}\varphi_{i}\left(z\right)\frac{\Psi (z)}{k_{B}T}\;\;\;a,b\in A$$

Here, $\chi_{a,b}$ is the Flory–Huggins parameter that describes hydrophobic interactions (in our case between polyelectrolyte chains and the water molecules); $\varphi^{bulk}$ is the volume fraction of components in bulk solution; *e* is the elementary charge; $z_{i}$ is the dimensionless charge of a particle of type *i*; and $\Psi \left(z\right)=e\psi \left(z\right)/\left(k_{B}T\right)$, where ψ(z) is electrostatic potential given by the Poisson-Boltzmann equation: (22)$$\epsilon \epsilon_{0}\frac{\partial^{2}\psi \left(z\right)}{\partial z^{2}}=-q\left(z\right),$$
where ϵ≈80 is the relative dielectric constant of water, $\epsilon_{0}$ is the vacuum permittivity, and q(z) is the distribution of the total charge in each layer at a distance *z* from the grafting surface.

Based on the condition of minimization of the functional $\delta F/\delta \varphi_{i}=0$, the potential $u_{i}\left(z\right)$ can be represented in the form of three terms:(23)$$u_{i}\left(z\right)=\lambda \left(z\right)+\frac{\delta F_{int}}{\delta \varphi_{i}}=\alpha \left(z\right)+u_{i}^{FH}\left(z\right)+u_{i}^{C}\left(z\right)$$
the Lagrange field λ(z), the field of hydrophobic forces leading to the collapse of polyelectrolyte chains
(24)$$u_{a}^{FH}\left(z\right)=\frac{1}{2}\chi_{a,b}\sum_{i\in A,i\neq a}\left(\langle \varphi_{i}\left(z\right)\rangle -\varphi_{i}^{bulk}\right)$$
and the electrostatic field promoting swelling of charged chains:(25)$$u_{i}^{C}\left(z\right)=\alpha_{i}\Psi \left(z\right)+\epsilon \epsilon_{0}\frac{1}{2}\frac{\partial^{2}\psi \left(z\right)}{\partial z^{2}}$$

From another condition for minimizing the functional $\delta F/\delta u_{i}=0$ follows a method for calculating the distribution of the volume fraction of components:(26)$$\varphi_{i}\left(z\right)=\frac{\partial \left(-\ln Q[u_{i}\left(z\right)]\right)}{\partial u_{i}\left(z\right)}$$

The numerical solution of this equation is carried out using a special iterative procedure that uses special propagator matrices to calculate the statistical sums of polymer chains on a cubic lattice.

Within the framework of the SF-SCF method, a random initial field $u_{i}\left(z\right)$ is first specified, then the partition function of the grafted chains in this field is calculated, the volume fraction distributions $\varphi_{i}\left(z\right)$ are calculated from the partition function, and the initially specified field $u_{i}\left(z\right)$ is corrected according to these distributions $\varphi_{i}\left(z\right)$. The calculation loop continues until $u_{i}\left(z\right)$ and $\varphi_{i}\left(z\right)$ are consistent with each other for the specified structural parameters of the molecular system and the incompressibility condition. In this case, the distribution of the total charge q(z) and the distribution of the electrostatic potential Ψ(z) also become consistent. The algorithm of the procedure used in this work is described in detail in the work [36] cited above.

## 4. Conclusions

By using the SCF approach, the salt-controlled vertical segregation of the flat mixed polymer brushes containing polyelectrolyte and neutral chains was studied. It was assumed that the selectivity to the solvent was varied for polyelectrolyte chains while the others polymer chains in the brush remained hydrophilic and neutral. Solvent selectivity (i.e., the hydrophobicity of the polyelectrolyte chains) can be controlled by changing the temperature. At low salt concentrations, the polyelectrolyte chains swell and occupy the surface of the mixed brush. At high salt concentrations, the hydrophobic polyelectrolyte chains collapse and give place to neutral chains on the surface. By changing the selectivity of the solvent and the ionic strength of the solution, the surface properties of such mixed brushes can be controlled. Based on the numerical simulations results, it is shown how the critical selectivity corresponding to the segregation transition in polyelectrolyte/neutral brushes depends on the ionic strength of the solution. It is shown that at the same ionic strength, the critical selectivity increases with increasing degree of dissociation of charged groups, as well as with the increasing fraction of polyelectrolyte chains in the mixed brush. At low ionic strengths, the critical selectivity of the solvent decreases with increasing grafting density; at high ionic strengths, on the contrary, it increases. Within the framework of the mean field theory, a two-parameter model has been constructed that quantitatively describes these dependencies.

## Figures and Tables

**Figure 1 ijms-25-13175-f001:**
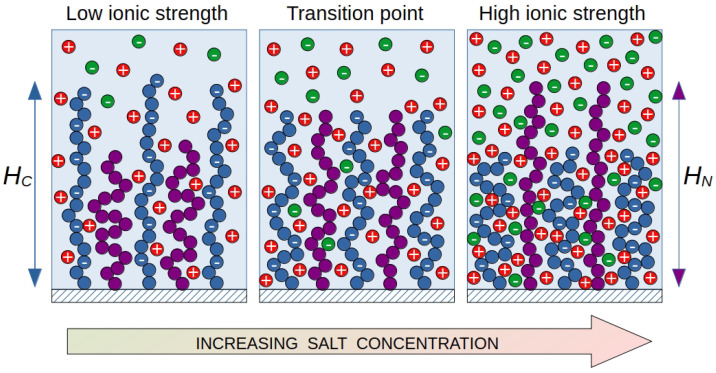
Schematic representation of vertical phase separation in a binary polymer brush consisting of neutral (purple circles) and partially charged polymer chains (blue circles) immersed into a monovalent salt solution.

**Figure 2 ijms-25-13175-f002:**
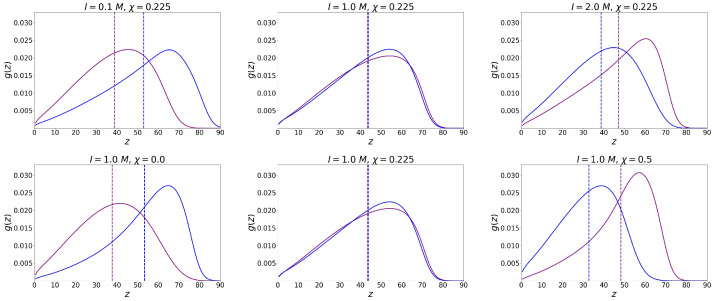
The ends distributions of polyelectrolyte (blue) and neutral (purple) chains. The total grafting density remains constant σ=0.1, the fraction of charged chains is X=0.5, and the polymerization degree of both types of chains is N=200. The vertical dotted lines represent the first moments of the distributions.

**Figure 3 ijms-25-13175-f003:**
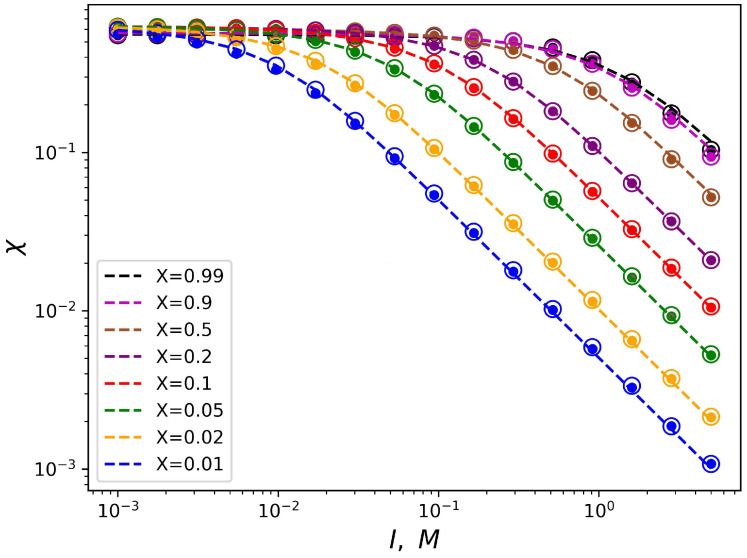
Dependencies of critical selectivity of the solvent on the ionic strength of the solution. The fraction of the polyelectrolyte chains in brush *X* varies. Small circles correspond to N=200, and large hollow symbols correspond to N=400. Dotted curves are approximation data using a two-parameter model (Equation (Equation 14)). The parameter area under the curves corresponds to the state when neutral chains are covered on top by charged ones. The remaining structural parameters are constant (α=0.1, σ=0.1).

**Figure 4 ijms-25-13175-f004:**
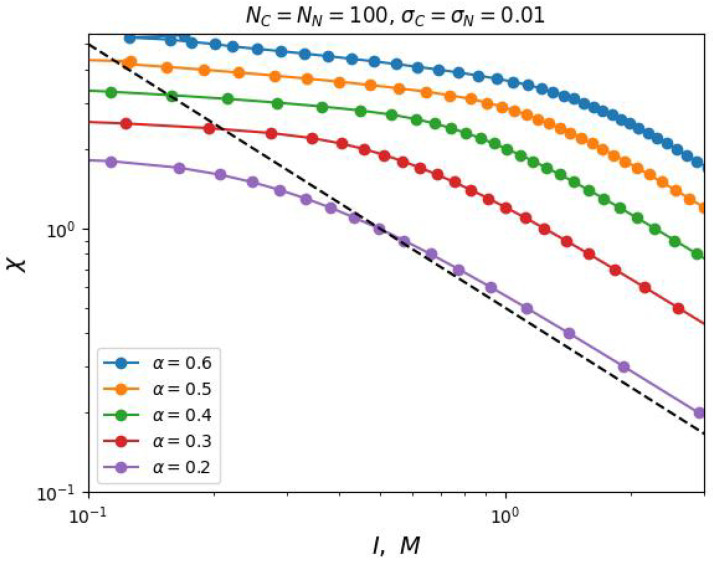
The value of critical selectivity, χ, (Flory–Huggins parameter of the interaction between a charged monomer unit and a solvent), as a function of the ionic strength, *I*, of a monovalent salt solution. The curves differ in the degree of ionization α of the polyelectrolyte chains. The graphs are plotted in double logarithmic coordinates, and the dashed line indicates slope −1.

**Figure 5 ijms-25-13175-f005:**
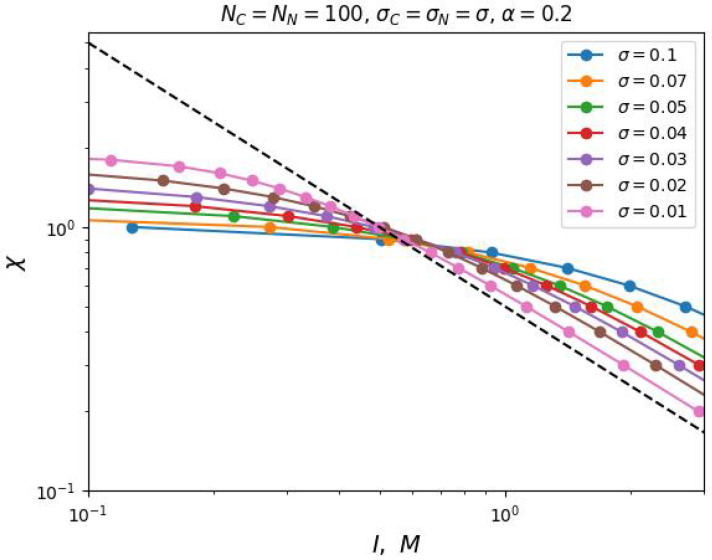
Critical selectivity value, χ, as a function of ionic strength, *I*, of the solution at different total grafting densities σ of both chain types. The graphs are plotted in double logarithmic coordinates, and the dashed line indicates slope −1.

## Data Availability

Data are contained within the article.

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
