# Peer review of "Salt-Controlled Vertical Segregation of Mixed Polymer Brushes"

_ijms, 2024, doi:10.3390/ijms252313175_

Round 1
Reviewer 1 Report
Comments and Suggestions for Authors
In this manuscript, the authors have done SCF computations to investigate the
influences of ionic strength (charge concentration) and the selectivity to solvent on
the segregation of mixed (charged plus neutral) polymer brushes. The results
presented are worth considering, however, I have some concerns, before I can
recommend for publication:
1-The term selectivity is referred to in different parts of the paper somehow
differently. In the abstract and in some places in the text it is written that the
selectivity is controlled by changing the temperature. In some other parts it is
defined in terms of the Florry-Huggins solubility parameter. In the abstract the
selectivity is also defined as “the hydrophobicity of the polyelectrolyte chains”.
Please address in the abstract/text the definition of the selectivity adopted in this
paper. It is true that the selectivity depends on some parameters as temperature. This
can be indicated in the introduction, in such a way that the readers do not get
confused with how the selectivity is defined in this paper.
2-While the system studied in this work is a polymer brush (charged plus neutral
polymer chains), in the abstract/text the term “solvent” is used for the polymer brush
interchangeably (for example in the abstract “Solvent selectivity (i.e., the
hydrophobicity of the polyelectrolyte chains) can be controlled …). May be the text
the authors have written is misleading in this seance. Please clearly define these
terms in the text (also in the caption of Figure 1, where the system is shown
schematically).
3-The term “share” has been used in some places the manuscript (probably to give
address to the concentration or mole fraction of polyelectrolyte in the brush). If so,
please clearly address to the concentration and the concentration scale (mole
fraction, volume fraction, or whatever scale it is used) adopted, and avoid using the
term “share”.
4-In the caption of Figure 1, please clearly indicate which type of chains are charged
and which are uncharged. It would also be helpful, if the same colour code (for
charged and uncharged chains) is used in Figure 2.
5-Figure 5 shows that at high grafting densities the selectivity less depends on the
ionic strength, than in low grafting densities. How do the authors interpret this
effect?
6-In the text and in Figure 5, please indicate the unit/dimension of grafting density.
7-Phase ordering in soft matter systems by tuning the ionic strength of the solution
to make a balance between hydrophobic and hydrophilic interactions is well-known
in the context of colloid chemistry. Although the structures formed are different
from what is presented in this paper, but the same lines of reasoning and
interpretations applies there. I would offer the authors (in case they think this may
add to the value of their paper) to give address to the self-assembly in patchy
colloidal particles by tuning the ionic strength of the solution, which has been
studied both experimentally (Nature, 2011, 469, 381) and computationally (Small,
2024, 20, 2306337).
8-Figure 5, please give address to the dashed line in the caption of the figure.
9-Figure 2: Labeling of the axis and the information given in the figure are not
readable. Please update this figure.
10-Figure 4: dotted line → dashed line
Comments on the Quality of English LanguageSome grammatical mistakes are seen in the text. The English of the manuscript needs improvement.
Author Response
The authors of the article are very grateful to the review for constructive remarks and comments. Next, we reply to comments in list items.
Comment 1: The term selectivity is referred to in different parts of the paper somehowdifferently. In the abstract and in some places in the text it is written that the selectivity is controlled by changing the temperature. In some other parts it is defined in terms of the Florry-Huggins solubility parameter. In the abstract the selectivity is also defined as “the hydrophobicity of the polyelectrolyte chains”.
Please address in the abstract/text the definition of the selectivity adopted in this paper. It is true that the selectivity depends on some parameters as temperature. This can be indicated in the introduction, in such a way that the readers do not get confused with how the selectivity is defined in this paper.
Response 1: For better understanding of the term "solvent selectivity", the beginning of the abstract has been rewritten as follows: «Using the self-consistent field approach, we studied the salt-controlled vertical segregation of mixed polymer brushes immersed into a selective solvent. We considered brushes containing two types of chains: polyelectrolyte (charged) chains and neutral chains. The hydrophobicity of both types of chains is characterized by the Flory-Huggins parameters χC and χN, respectively. It was assumed that the hydrophobicity is varied only for the polyelectrolyte chains (χC), while other polymer chains in the brush remain hydrophilic (χN=0) and neutral. Thus, in our model, the solvent selectivity (χ = χC - χN) was varied, which can be controlled in a real experiment, for example, by changing the temperature.»
The last paragraph in the Method subsection has been corrected and supplemented:
«Neutral and charged chains also have different affinities for the solvent, i.e. the solvent is selective with respect to different types of grafted chains. The hydrophobicity of the polyelectrolyte chains and the neutral chains in the brush is characterized by the Flory-Huggins parameters, χC and χN respectively. It is assumed that the solvent is athermal with respect to the neutral chains (χN=0), but polyelectrolyte chains are capable of changing their hydrophobicity. The strength of selectivity is specified in the model as the difference χ = χC - χN. The same definition of selectivity is used, for example, by the authors of the article \cite{Romeis2015}. Since χN=0, the solvent selectivity is characterized by the incompatibility of water molecules and monomer units of charged chains.»
Comment 2: While the system studied in this work is a polymer brush (charged plus neutral polymer chains), in the abstract/text the term “solvent” is used for the polymer brush interchangeably (for example in the abstract “Solvent selectivity (i.e., the hydrophobicity of the polyelectrolyte chains) can be controlled …). May be the text the authors have written is misleading in this seance. Please clearly define these terms in the text (also in the caption of Figure 1, where the system is shown schematically).
Response 2: The polyelectrolyte and neutral chains are immersed into an aqueous solution of a monovalent salt, and this solution is the solvent for the polymer brush. This is described in the Method subsection. The relationship between the hydrophobicity of the charged chains and the selectivity of the solvent is explained in the previous response.
Comment 3: The term “share” has been used in some places the manuscript (probably to give address to the concentration or mole fraction of polyelectrolyte in the brush). If so, please clearly address to the concentration and the concentration scale (mole fraction, volume fraction, or whatever scale it is used) adopted, and avoid using the term “share”.
Response 3: At the beginning of the Method subsection, an explanatory sentence is inserted: «The molar fraction of polyelectrolyte chains X = σC /(σC + σN) is the ratio of the number of polyelectrolyte chains to the total number of chains in the brush.» Throughout the text, the word “share” is replaced by “fraction”.
Comment 4: In the caption of Figure 1, please clearly indicate which type of chains are charged and which are uncharged. It would also be helpful, if the same colour code (for charged and uncharged chains) is used in Figure 2.
Response 4: In the caption of Figure 1, an explanation of the color scheme for neutral and charged circuits has been added. The same color code was used in the new version of Figure 2.
Comment 5: Figure 5 shows that at high grafting densities the selectivity less depends on the ionic strength, than in low grafting densities. How do the authors interpret this effect?
Response 5: This effect was shown in our article for the first time and is a subject for discussion. It can be argued that in the limit of high grafting densities (when the total grafting density is equal to one) the volume of the brush will not contain solvent and all chains will be maximally stretched to the same limiting height. Under these conditions, segregation transition becomes impossible at any ionic strength of the solution and at any selectivity of the solvent.
Comment 6: In the text and in Figure 5, please indicate the unit/dimension of grafting density.
Response 6: The grafting density is a dimensionless quantity. The following definition has been added to the Method subsection: « The grafting density is a dimensionless quantity equal to the ratio σ = a2/s, where a is the linear size of the monomer unit of the grafted chain, s is the grafting surface area per grafted macromolecule. In other words, the grafting density σ is the number of chains per unit area of the grafting surface.»
Comment 7: Phase ordering in soft matter systems by tuning the ionic strength of the solution to make a balance between hydrophobic and hydrophilic interactions is well-known in the context of colloid chemistry. Although the structures formed are different from what is presented in this paper, but the same lines of reasoning and interpretations applies there. I would offer the authors (in case they think this may add to the value of their paper) to give address to the self-assembly in patchy colloidal particles by tuning the ionic strength of the solution, which has been studied both experimentally (Nature, 2011, 469, 381) and computationally (Small, 2024, 20, 2306337).
Response 7: We thank the reviewer for providing references to useful articles, we will definitely use them in our future works.
Comment 8: Figure 5, please give address to the dashed line in the caption of the figure.
Response 8: The following text has been added to the caption of Figure 5: « The graphs are plotted in double logarithmic coordinates, the dashed line indicates slope -1.»
Comment 9: Figure 2: Labeling of the axis and the information given in the figure are not
readable. Please update this figure.
Response 9: The axis markings and information shown in Figure 2 have been made larger.
Comment 10: Figure 4: dotted line → dashed line
Response 10: This typo in caption to Figure 4 is corrected.
Reviewer 2 Report
Comments and Suggestions for Authors
The authors investigated the surface-switching properties of polyelectrolyte/neutral mixed brushes on a flat surface in a salt solution, utilizing the one-gradient Scheutjens-Fleer Self-Consistent Field (SF-SCF) method. They identified the force balance in the system perpendicular to the brush layer, driven by (1) osmotic pressure, (2) elastic pressure from chain stretching, (3) mixing entropy between the polymer and solvent, and (4) pressure from hydrophobic compression of polyelectrolyte chains in a selective solvent. Using these analytical formulations, the authors developed a two-coefficient model that governs polymer chain behavior under these conditions.
Additionally, they examined the model’s limitations at high concentrations, attributing observed deviations to the non-uniform distribution of monomer density and charge within the brush phase, which complicates the simplified description of the “switching effect.” This paper is well-prepared, with clear formulations and derivations, and I recommend it for publication in IJMS in its current form.
Author Response
Comment 1: The authors investigated the surface-switching properties of polyelectrolyte/neutral mixed brushes on a flat surface in a salt solution, utilizing the one-gradient Scheutjens-Fleer Self-Consistent Field (SF-SCF) method. They identified the force balance in the system perpendicular to the brush layer, driven by (1) osmotic pressure, (2) elastic pressure from chain stretching, (3) mixing entropy between the polymer and solvent, and (4) pressure from hydrophobic compression of polyelectrolyte chains in a selective solvent. Using these analytical formulations, the authors developed a two-coefficient model that governs polymer chain behavior under these conditions.
Additionally, they examined the model’s limitations at high concentrations, attributing observed deviations to the non-uniform distribution of monomer density and charge within the brush phase, which complicates the simplified description of the “switching effect.” This paper is well-prepared, with clear formulations and derivations, and I recommend it for publication in IJMS in its current form.
Response 1: The authors of the article express their deep gratitude to the reviewer for the positive review.
Reviewer 3 Report
Comments and Suggestions for Authors
Summary: In this paper, the simple box-model and the numerical SCF model are employed to investigate the effect of ionic strength, chain grafting density, the degree of ionization of polyelectrolyte chains, and the percentage of polyelectrolyte chains within the grafted polymer layer on the critical selectivity of mixed polymer brushes. This critical selectivity is important since it indicates the transition point where one type of grafted polymer chain is about to be shielded by the other, and therefore the start of the change of surface properties. The Reviewer thinks this manuscript is well-written and recommends it to be published after some slight changes.
Comments:
1. p4, line 110. Since hydrophobicity is used throughout the whole paper, would it be possible to replace “solvatophobicity” with “hydrophobicity”?
2. p4, line 114. Why do the authors put “(a,b)” here? What does this refer to?
3. This is a general comment. The reviewer noticed that the authors use both “Figure x” and “figure x” in this manuscript. Please be consistent.
4. p5, equation (12). “N” is used here but lacks the definition. Please consider adding this on line 147.
5. p7, between equation (23) and (24). The authors used “χ*” already to refer to the critical χ. Please be consistent and avoid using “chi*”.
6. p.8, Figure 3. It seems like the colors of solid dots representing χ = 0.5, 0.9, and 0.99 are not in line with the hollow dots as well as the dash lines for the same χ. May the authors consider aligning them?
7. p.10, line 222. It says that “the critical selectivity increases with increasing chain grafting density”, which is only true in the electrolyte with high ionic strength, and this trend is flipped in the low ionic strength according to Figure 5. It is suggested that the authors may want to be more specific about under which ionic strength conditions, the critical selectivity starts to increase or decrease along with the change of the grafting density. Please consider adding more details for better clarification.
Author Response
The authors of the article are very grateful to the review for constructive remarks and comments. Next, we reply to comments in list items.
Comment 1: p4, line 110. Since hydrophobicity is used throughout the whole paper, would it be possible to replace “solvatophobicity” with “hydrophobicity”?
Response 1: The word "solvatophobicity" has been replaced by "hydrophobicity"
Comment 2: p4, line 114. Why do the authors put “(a,b)” here? What does this refer to?
Response 2: This typo has been corrected: «(a,b)» → «(I, χ)»
Comment 3: This is a general comment. The reviewer noticed that the authors use both “Figure x” and “figure x” in this manuscript. Please be consistent.
Response 3: Throughout the text, the word "Figure" is now used only with a capital letter after article correction.
Comment 4: p5, equation (12). “N” is used here but lacks the definition. Please consider adding this on line 147.
Response 4: An explanation has been added after equation (12): «N is the degree of polymerization of these chains»
Comment 5: p7, between equation (23) and (24). The authors used “χ*” already to refer to the critical χ. Please be consistent and avoid using “chi*”.
Response 5: This typo has been corrected.
Comment 6: p.8, Figure 3. It seems like the colors of solid dots representing χ = 0.5, 0.9, and 0.99 are not in line with the hollow dots as well as the dash lines for the same χ. May the authors consider aligning them?
Response 6: The color map alignment in Figure 3 was performed.
Comment 7: p.10, line 222. It says that “the critical selectivity increases with increasing chain grafting density”, which is only true in the electrolyte with high ionic strength, and this trend is flipped in the low ionic strength according to Figure 5. It is suggested that the authors may want to be more specific about under which ionic strength conditions, the critical selectivity starts to increase or decrease along with the change of the grafting density. Please consider adding more details for better clarification.
Response 7: The following clarification has been added to the text: «At low ionic strengths, the critical selectivity of the solvent decreases with increasing grafting density; at high ionic strengths, on the contrary, it increases.» The corresponding changes have also been made to the abstract of the article.
Round 2
Reviewer 1 Report
Comments and Suggestions for Authors
I am satisfied with the revisions the authors did.